# Effect of Maternal Glucose and Triglyceride Levels during Early Pregnancy on Pregnancy Outcomes: A Retrospective Cohort Study

**DOI:** 10.3390/nu14163295

**Published:** 2022-08-11

**Authors:** Dandan Wu, Jianlin Zhang, Yimeng Xiong, Hui Wang, Danyang Lu, Mengxi Guo, Jian Zhang, Lei Chen, Jianxia Fan, Hefeng Huang, Xianhua Lin

**Affiliations:** 1The International Peace Maternity and Child Health Hospital, School of Medicine, Shanghai Jiao Tong University, Shanghai 200030, China; 2Obstetrics and Gynecology Hospital, Institute of Reproduction and Development, Fudan University, Shanghai 200090, China; 3Department of Obstetrics and Gynecology, Maternity and Child Health Hospital of Songjiang District, Shanghai 201620, China; 4Research Units of Embryo Original Diseases, Chinese Academy of Medical Sciences, Shanghai 201203, China; 5Key Laboratory of Reproductive Genetics, Ministry of Education, Department of Reproductive Endocrinology, Women’s Hospital, Zhejiang University School of Medicine, Hangzhou 310006, China; 6Shanghai Key Laboratory of Embryo Original Diseases, Shanghai 200030, China

**Keywords:** gestational diabetes mellitus, hypertensive disorders in pregnancy, maternal dysglcemia, maternal triglyceride, maternal and infant outcomes

## Abstract

Maternal dysglycemia and lipid metabolic dysfunction have been recognized as risk factors for pregnancy complications and adverse perinatal outcome jointly and separately, but current diagnostic window-period which is at the end of the second trimester might be late to avoid chronic adverse impacts on both mother and fetus. A retrospective cohort study involving 48,973 women with fasting blood glucose (FPG) below diagnostic thresholds and lipid screening in early pregnancy was performed. Data of pregnancy outcomes including gestational diabetes mellitus (GDM), hypertensive disorders in pregnancy (HDP), and neonatal outcomes were obtained for multivariable logistic analysis. As a result, higher FPG (≥75th, 4.68 mM) significantly increased risks of GDM (Adjusted odds ratio (AOR), 2.81; 95% CI, 2.60 to 3.05) and HDP (1.98; 1.81 to 2.16), and slightly increased risks of large for gestational age (LGA), macrosomia births and neonatal intensive care unit (NICU) compared to women with low FPG (≤25th, 4.21 mM). High maternal triglyceride (mTG) level had higher risks of GDM and HDP in all maternal FPG strata. Further analysis showed that women of top quartile of glucose combined with upper 10 percentile triglyceride have higher risks for GDM (AOR, 5.97; 95% CI, 5.26 to 6.78; risk difference 30.8, 95% CI 29.2 to 32.3) and HDP (AOR, 2.56; 95% CI, 2.20 to 2.99, risk difference 11.3, 95% CI 9.9 to 12.7) when compared to those in women of the bottom strata after adjustment. Therefore, both the early-pregnancy FPG and mTG levels should be screened among overall population including the low-risk population to reduce the incidence of pregnancy complications.

## 1. Introduction

Maternal dysglycemia during pregnancy has been recognized as a major modifiable risk factor for pregnancy complications and adverse perinatal outcomes, such as gestational diabetes, hypertensive disorders of pregnancy, postpartum hemorrhage, fetal overgrowth, and birth injury [1,2,3,4,5]. Owing to the change of lifestyle, increasing number of women developed abnormal glucose metabolism before and during pregnancy [6].

GDM is defined as diabetes diagnosed in the second or third trimester of pregnancy that excluding pre-gestational diabetes. Medical interventions, including glucose monitoring, diet and exercise modification and pharmacological therapies, were usually performed in women with overt diabetes or GDM diagnosed. The diagnostic window-period at the end of the second trimester might be late to avoid chronic adverse impacts on both mother and fetus, for early pregnancy been known as the critical period of embryonic and placental development [7]. Many professional associations, including the International Association of Diabetes and Pregnancy Study Groups (IADPSG), the American Diabetes Association (ADA), the International Federation of Gynecology and Obstetrics (FIGO), and the American College of Obstetricians and Gynecologists (ACOG), have recommended universal or selective screening for glucose levels in early pregnancy but the screening methods and standards lack uniformity. IADPSG recommends that fasting plasma glucose (FPG) tested at any time during pregnancy, in the range 5.1–6.9 mM, should be considered diagnostic of GDM, while screening for undiagnosed pre-existing prediabetes and diabetes in early pregnancy, is suggested only among high-risk population by ADA and ACOG [8,9,10,11]. Analogously, early screening for diabetes and GDM is not recommended by the Ministry of Health of China [12]. Considering the cost-effective and clinical directive significance, more work is needed to be conducted aiming to explore the effects of maternal glucose metabolism on pregnancy complications and adverse perinatal outcomes from first trimester onwards in women with FPG below the threshold of prediabetes and diabetes.

Gradual increases of lipid profiles, as characteristic metabolic features, are needed for fetal growth and development [13]. It was reported that the number of women with subclinical lipid metabolic dysfunction at reproductive age is increasing. Maternal triglyceride (mTG), which processed the transfer of maternal fatty acids to the fetus [14], was reported to be associated with fetal growth and pregnancy complications [15]. It was well studied that the abnormal lipid metabolism such as hypertriglyceridemia induced insulin resistance, which played a key role in the pathophysiology of maternal dysglycemia during pregnancy [16,17].

Thus, we conducted this study aiming to evaluate the associations of maternal glucose below diagnostic thresholds during early pregnancy, with the risks of pregnancy complications and adverse perinatal outcomes. We also aimed to assess the combined effect of maternal glucose and triglyceride in early pregnancy on the risks of pregnancy outcomes.

## 2. Materials and Methods

### 2.1. Study Population

This retrospective cohort study was conducted in International Peace Maternity and Child Health Hospital (IPMCH) in Shanghai, China from June 2013 to January 2018. All women with a baby delivered recorded in the electronic medical record system of the hospital were enrolled in this study. The clinical and demographic information were extracted from the medical record system under monitor of the research ethics committee at IPMCH. A total of 50,790 pregnant women were enrolled, all of whom underwent fasting blood glucose as well as lipid screening in early pregnancy and received regular systematic antenatal examination. The following exclusion criteria were used: (1) multiple pregnancies, (2) stillbirth, (3) pre-gestational diabetes or prediabetes, including undiagnosed pre-existing diabetes or prediabetes, (4) pre-pregnancy hypertension (SBP ≥ 140 mmHg or DBP ≥ 90 mm Hg), (5) renal disease. A total of 48,973 cases with FPG and triglyceride data in early pregnancy were further analyzed (Figure 1).

### 2.2. Data Collection

Maternal baseline characteristics, including age at delivery, parity, education, birthplace, medical history, were obtained at first prenatal visit. Prenatal smoking and alcohol use was extremely rare in the study population. Gestational age was calculated according to the last menstrual period and adjusted by the ultrasonic data in early pregnancy. Maternal FPG and serum TG concentrations were determined by fasting blood samples in early pregnancy, which were collected at the first prenatal visit between 7:00–9:00 a.m. during 9 to 14 weeks of gestation. The diagnoses of prediabetes and pre-gestational diabetes were based on the clinical records or reported by the participants. Women with FPG at the first visit, in the range 5.60–6.99 or ≥7 mM, were also diagnosed as prediabetes or pre-gestational diabetes. FPG values in early pregnancy were divided into four categories according to the quartiles (Q) of all the measurements below diagnostic thresholds. Values below the 25th percentile were defined as Q1, values between the 25th and 50th percentile were defined as Q2, values between the 50th and 75th percentile were defined as Q3, and values equal to or above 75th percentile were defined as Q4. “High mTG” was defined as values equal to or above the 90th percentile in early pregnancy. Maternal pre-pregnancy body mass index (BMI) was categorized as: underweight (BMI < 18.5 kg/m^2^), normal weight (BMI 18.5 to <25 kg/m^2^), overweight or obese (BMI ≥ 25 kg/m^2^) [18].

### 2.3. Outcomes

All women enrolled underwent a diagnostic 75-g oral glucose tolerance test (OGTT) at 24–28 weeks of gestation. The diagnosis of GDM was based on OGTT according to the IADPSG criteria (any of OGTT values were met or exceeded: fasting ≥ 5.1 mM; 1 h ≥ 10.0 mM; 2 h ≥ 8.5 mM) [11]. Hypertensive disorders in pregnancy (HDP) included pregnancy-induced hypertension (PIH) and preeclampsia. PIH was defined as systolic blood pressure (SBP) ≥ 140 mmHg or diastolic blood pressure (DBP) ≥ 90 mmHg for least twice, 4 h apart, after 20 weeks of gestation in a previously normotensive woman. Preeclampsia was defined by the combination of gestational hypertension and newly onset proteinuria or signs of organ dysfunction occurring after 20 weeks’ gestation. PIH analysis did not include preeclampsia cases and vice versa [19]. Intrahepatic cholestasis of pregnancy (ICP) was diagnosed as fasting serum bile acid concentrations ≥10 μmol/L in the third trimester of gestation.

Information of neonatal outcomes were obtained from birth records, including birth weight, preterm delivery (delivery before 37 completed weeks), small for gestational age (SGA) and large for gestational age (LGA) infant (defined as birthweight ≥90th and ≤10th percentile adjusted for the sex of neonatal and gestational age, respectively), low birth weight (<2500 g), macrosomia (≥4000 g), 10 min Apgar score, and admission to the neonatal intensive care unit (NICU) [20].

### 2.4. Statistical Analysis

Continuous variables were presented as means ± SD. The differences in the continuous variables between groups were tested by one-way ANOVA. Categorical variables were presented as frequencies with proportions. Logistic regressions were used to estimate odds ratios (ORs) with 95% CIs of pregnancy complications and perinatal outcomes in participants. Multivariable analysis was used to estimate separate and combined associations of FPG and mTG in early pregnancy, with the risks of pregnancy complications and outcomes. Potential confounders were considered in the multivariable logistic analysis. Covariates, education (≤9, 10–12, 13–15, or ≥16 years), birthplace (residents or immigrants), parity (once, or more than once), maternal age (≤24, 25–29, 30–34, or ≥35 years), were included. Different pre-pregnancy BMI ranges (BMI < 18.5, BMI 18.5 to <25, BMI ≥ 25 kg/m^2^) were also added to above-mentioned covariates when the analysis was conducted among the whole study cases.

Absolute risk difference compared to normal mTG and low FPG was calculated as the difference between the risk of any adverse outcome in the high mTG and high FPG group and the risk of that event in the normal mTG and low FPG group. The confidence interval was obtained with standard statistical packages [21]. A confidence interval that contained a zero meant that there was no significant difference between the event and the control in terms of risk.

SPSS version 16.0 (SPSS Inc., Chicago, IL, USA) was used for all statistical analyses. A two-tailed *p* value of 0.05 was used as the threshold for statistical significance.

## 3. Results

### 3.1. Baseline Characteristics of the Study Population

In the study, 48,973 women who were enrolled had a mean (SD) age of 30.42 (3.76) years. The cutoff points of FPG at the 25th (4.21 mM), 50th (4.44 mM) and 75th (4.68 mM) percentiles were defined according to the assayed values among women enrolled. The cutoff points of TG at the 10th (0.78 mM) and 90th (2.00 mM) percentiles were also defined according to the assayed values.

Among women enrolled, 10.5% were overweight or obese, 80.7% were nulliparous, 70.8% were highly educated, and 79.7% were residents in Shanghai (Table 1). Table 1 shows that FPG levels and TG levels were significantly higher in women of older age (≥35 years) or a higher pre-pregnancy BMI (≥25 kg/m^2^), who were multiparous, and had a lower education level. During the study period, 6655 (13.6%) women were diagnosed with GDM, 4787 (9.8%) women were diagnosed with HDP. Compared with mean levels of FPG (4.44 mM) and TG levels (1.34 mM) of all 48,973 women, mothers with GDM or HDP had a significantly higher mean FPG and TG levels in early pregnancy.

### 3.2. FPG Levels in Early Pregnancy and Risks of Pregnancy Outcomes

Compared to women with FPG values below the 25th percentile (Q1), risks of GDM, HDP, PIH, and preeclampsia increased with FPG levels (Table 2). The rates and adjusted ORs (AORs) of GDM were as follows: FPG 25th to less than 50th (10.6%; AOR, 1.27; 95% CI, 1.16 to 1.38), FPG 50th to less than 75th (13.1%; AOR, 1.56; 95% CI, 1.44 to 1.70), FPG of 75th or greater (22.4%; AOR, 2.81; 95% CI, 2.60 to 3.05). The rates and AORs for HDP were as follows: FPG 25th to less than 50th (7.2%; AOR, 1.08; 95% CI, 0.97 to 1.19), FPG 50th to less than 75th (11.9%; AOR, 1.84; 95% CI, 1.68 to 2.02), FPG of 75th or greater (13.3%; AOR, 1.98; 95% CI, 1.81 to 2.16). We also found that higher FPG levels increased the risks of PIH and preeclampsia before and after adjustment for pre-pregnancy BMI, years of education, birthplace, parity, and maternal age. However, no statistically significant risks of ICP and postpartum hemorrhage were found. Compared with low values of FPG, women with Q2, Q3, Q4 values of FPG slightly increased the risks of LGA and macrosomia births and NICU (Table 2).

### 3.3. FPG Levels and TG Levels in Early Pregnancy and Risks of Pregnancy Outcomes

To determine whether mTG levels increased risks of GDM and HDP in all FPG strata in early pregnancy, we analyzed the risks of main pregnancy outcomes among women stratified by early-pregnancy FPG levels in conjunction with mTG levels. As compared to low values of mTG level, higher mTG level had higher risks of GDM and HDP in all maternal FPG strata (Table 3): FPG less than 25th (GDM: AOR, 2.53; 95% CI, 2.12 to 3.02; HDP: AOR, 1.57; 95% CI, 1.26 to 1.96, respectively), FPG 25th to less than 50th (GDM: AOR, 2.10; 95% CI, 1.78 to 2.48; HDP: AOR, 1.74; 95% CI, 1.43 to 2.13, respectively), FPG 50th to less than 75th (GDM: AOR, 1.77; 95% CI,1,52 to 2.06; HDP: AOR, 1.34; 95% CI, 1.13 to 1.59, respectively), and FPG 75th or greater (GDM: AOR, 2.07; 95% CI, 1.85 to 2.32; HDP: AOR, 1.28; 95% CI, 1.10 to 1.48, respectively).

Then we further analyzed the combined effect of FPG and TG levels on these pregnancy outcomes (Figure 2). The results showed that, as compared to mothers with low values of FPG level and normal TG levels, Q2, Q3, Q4 values with high TG levels mothers had higher risks of GDM and HDP, PIH and preeclampsia after adjustment for pre-pregnancy BMI, years of education, birthplace, parity, and maternal age. Risks of any pregnancy complications increased with FPG values. The highest risks of GDM and HDP were observed in mothers with Q4 value FPG and high TG levels as follows (among mothers with upper 10% mTG levels): FPG less than 25th (GDM: 18.7%; AOR, 2.53; 95% CI, 2.12 to 3.02; risk difference 11.5, 95%CI, 9.8 to 13.5; HDP: 10.7%; AOR, 1.60; 95% CI, 1.29 to 1.98; risk difference 4.5, 95%CI 2.9 to 6.0, respectively), FPG 25th to less than 50th (GDM: 20.9%; AOR, 2.71; 95% CI, 2.30 to 3.20; risk difference 13.8, 95%CI 12.1 to 15.5; HDP: 12.7; AOR, 1.85; 95% CI, 1.52 to 2.51, risk difference 6.5, 95%CI 4.9 to 8.0, respectively), FPG 50th to less than 75th (GDM: 22.4%; AOR, 2.91; 95% CI, 2.49 to 3.40; risk difference 15.2, 95%CI 13.6 to 16.8; HDP: 16.2%; AOR, 2.51; 95% CI, 2.11 to 2.99, risk difference 10.0, 95%CI 8.5 to 11.5, respectively), and FPG 75th or greater (GDM: 37.9%; AOR, 5.97; 95% CI, 5.26 to 6.78; risk difference 30.8, 95%CI 29.2 to 32.3; HDP: 17.5%; AOR, 2.56; 95% CI, 2.20 to 2.99, risk difference 11.3, 95%CI 9.9 to 12.7, respectively). Similarly increased risks of PIH and preeclampsia with FPG values were observed among women with high TG levels. Mothers in Q2, Q3, Q4 values of FPG were also, with normal TG mothers, associated with a higher risk of any pregnancy complication (Figure 2).

All analyses were adjusted for age, education, parity, place of birth, and pre-pregnancy BMI. FPG values in early pregnancy were divided into four categories according to the quartiles (Q): FPG Q1: FPG < 25th; FPG Q2: 25th ≤ FPG < 50th; FPG Q3: 50th ≤ FPG < 75th; FPG Q4: FPG ≥ 75th. “High mTG” was defined as values equal to or above the 90th percentile in early pregnancy. Maternal pre-pregnancy BMI was categorized as: underweight (BMI < 18.5 kg/m^2^), normal weight (BMI 18.5 to < 25 kg/m^2^), overweight or obese (BMI ≥ 25 kg/m^2^). Adjusted odds ratios (AOR) and risk differences of GDM, HDP, PIH and preeclampsia for different FPG and mTG categories are calculated by comparing with the reference group (FPG < 25th and mTG< 2.00 mM). All risk differences, adjusted odds ratios, and each corresponding 95% CIs were calculated from the results of multivariable models and adjusted for baseline risk imprecision.

## 4. Discussion

In the present study, maternal high early-pregnancy FPG, below diagnostic thresholds, had higher risks of GDM and HDP. The incidence of GDM in women with high FPG (FPG ≥ 75th, 4.68 mM) was 22.4%, which is nearly 3 times higher than that in the bottom quartile of FPG (8.2%). In addition, the incidences of HDP also increased from 6.6% to 13.3% as the FPG elevated, though below the diagnostic threshold. Stratified by the quartiles of maternal FPG, the risks of GDM and HDP for women with TG ≥ 90th during early-pregnancy, increased in all maternal FPG strata compared with those of TG < 90th. The risk of GDM among women in high mTG group of the top quartile of FPG was about almost 6 times higher than those in the bottom tranche, and the risk for HDP was higher than twice as well.

As one of the most common pregnancy complications, GDM increases risks of pregnancy complications of both mother and fetus. Moreover, women who were diagnosed as GDM, along with their offspring, were identified to have higher risks of type 2 diabetes, premature cardiovascular disease in long term. Meanwhile, HDP, affecting 2–10% pregnancies, as a dangerous complication, may result in adverse consequences for mothers and offspring, when it proceeds without supervision and administration [22,23].

Detecting risk factors for pregnancy complications in early phase might be of great value in improving pregnancy outcomes. Several RCTs studied the effect of medical intervention, including lifestyle modification and pharmacological therapies, on the risks of GDM and attendant pregnancy outcomes, and came to inconsistent results: some studies suggested the intervention should be considered early in pregnancy [24]. A meta-analysis indicated that lifestyle modification (diet, physical activity, or both) initiated before 15 weeks of gestation, but not afterwards, can reduce the risk of GDM [25].

Whether early glucose screening should be recommended at the first prenatal visit for all women or confine to those with specific risk factors remains inconsistent. ADA recommended that testing should be considered in overweight or obese women (BMI ≥ 25 or 23 kg/m^2^ in Asian populations) who have one or more risk factors, such as prior GDM, a family history of diabetes, polycystic ovary syndrome, cardiovascular diseases or hypertension, a triglyceride level ≥ 250 mg/dL (2.82 mM) [8]. However, the proportion of pregnancy women with a triglyceride level ≥ 2.82 mM, was only about 2.8% in our overall cohort. Some studies reported that FPG in early pregnancy was a poor predictor for GDM in late pregnancy with a low sensitivity or poor specificity, while other studies suggested women with FPG ≥ 5.60 mM should be provided with nutrition and exercise advice in Chinese population [24,26,27]. In recent studies, women with the higher FPG levels across all ranges in early pregnancy were reported to have increased risks of delivering babies of LGA and macrosomia [28] while FPG below the diagnostic threshold and SGA were inversely related [29]. Consistently, we also observed slightly increased risk of fetal overgrowth (both LGA and macrosomia) and decreased risk of SGA in mother with higher early-pregnancy FPG levels below the threshold. Moreover, higher rates of GDM and HDP were identified in women with gradually elevated FPG level (below diagnostic thresholds).

Furthermore, elevated maternal TG is regarded as an essential moderating variable of insulin resistance, frequently accompanied with the processes of dysglycemia. Our data indicated high mTG in all maternal FPG strata contributed to significantly elevated risks of GDM and HDPs (both PIH and preeclampsia). Since maternal TG levels began to vary from the end of first trimester, its level in early pregnancy reflects the baseline lipid metabolism of mother. Some studies reported elevated mTG in middle and late pregnancy was correlated with fetal growth in GDM [30]. The association between increased risk of PE and mTG levels (but not total cholesterol levels) in early pregnancy, were found in our previous study [31].

## 5. Conclusions

Elevated maternal fasting blood glucose level at early pregnancy, in the range below diagnostic thresholds, was associated with significantly increased risks of GDM and HDP, and slightly increased risks of preterm delivery, LGA and NICU. Maternal high FPG combined with high triglyceride level at early pregnancy could have higher risks of GDM and HDP than women of lower strata with normal mTG levels after adjustment for confounding factors. Therefore, both the early-pregnancy FPG and mTG levels should be screened for etiological and preventive perspective among overall population including the low-risk population. Future preventive interventions should be focused on monitoring and managing blood glucose level across all ranges from early pregnancy, and the lipid level especially TG level to reduce the incidence of pregnancy complications.

## Figures and Tables

**Figure 1 nutrients-14-03295-f001:**
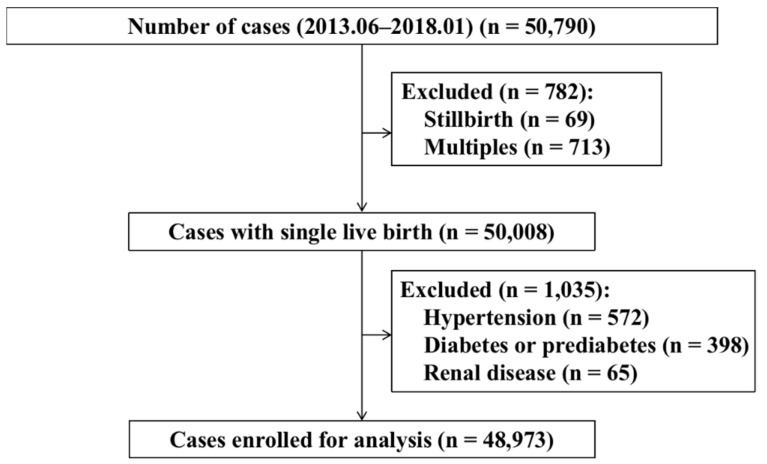
Flowchart for the study population.

**Figure 2 nutrients-14-03295-f002:**
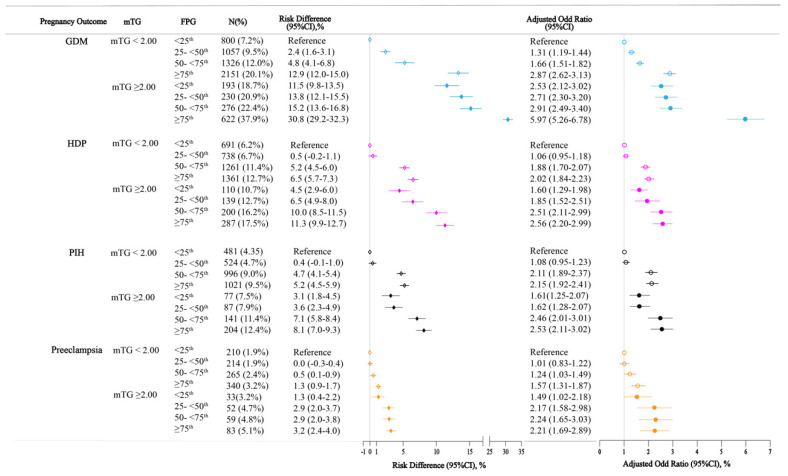
Combined effects of early-pregnancy FPG and mTG on pregnancy complications.

**Table 1 nutrients-14-03295-t001:** Maternal characteristics and pregnancy complications baseline data of the study population.

Characteristics	*n* (%)	FPG Level (mM)	*p* Value	mTG Level (mM)	*p* Value
Total	48,973	4.44 ± 0.36		1.34 ± 0.55	
Age, years			0.000		0.000
≤24	1518 (3.1%)	4.40 ± 0.35		1.20 ± 0.45	
25–29	15,043 (30.7%)	4.41 ± 0.36		1.27 ± 0.49	
30–34	25,076 (51.2%)	4.45 ± 0.36		1.35 ± 0.56	
≥35	7336 (15.0%)	4.51 ± 0.37		1.48 ± 0.61	
Pre-pregnancy BMI, kg/m^2^			0.000		0.000
<18.5	5643 (11.5%)	4.38 ± 0.36		1.15 ± 0.39	
18.5–25	38,191 (78.0%)	4.43 ± 0.36		1.33 ± 0.53	
≥25	5139 (10.5%)	4.56 ± 0.38		1.62 ± 0.69	
Parity, n			0.000		0.000
0	39,520 (80.7%)	4.43 ± 0.36		1.31 ± 0.53	
≥1	9453 (19.3%)	4.50 ± 0.37		1.46 ± 0.60	
Education, years			0.000		0.000
9	843 (1.7%)	4.51 ± 0.38		1.51 ± 0.68	
10–12	2650 (5.5%)	4.50 ± 0.36		1.41 ± 0.60	
13–15	10,665 (22.0%)	4.47 ± 0.36		1.36 ± 0.57	
≥16	34,350 (70.8%)	4.43 ± 0.36		1.32 ± 0.53	
Birthplace			0.903		0.000
Residents	39,020 (79.7%)	4.44 ± 0.36		1.33 ± 0.54	
Immigrants	9953 (20.3%)	4.44 ± 0.37		1.37 ± 0.57	
Category by FPG in early pregnancy			0.000		0.000
<25th percentile	12,175	3.99 ± 0.19		1.30 ± 0.51	
25– < 50th percentile	12,181	4.33 ± 0.07		1.31 ± 0.53	
50– < 75th percentile	12,260	4.55 ± 0.07		1.34 ± 0.54	
≥75th percentile	12,357	4.90 ± 0.19		1.41 ± 0.61	
Category by mTG in early pregnancy			0.000		0.000
<90th percentile	43,970	4.43 ± 0.36		1.20 ± 0.34	
≥90th percentile	5003	4.51 ± 0.39		2.52 ± 0.62	
GDM			0.000		0.000
No	42,318 (86.4%)	4.42 ± 0.35		1.30 ± 0.52	
Yes	6655 (13.6%)	4.60 ± 0.39		1.57 ± 0.67	
HDP			0.000		0.000
No	44,186 (90.2)	4.43 ± 0.36		1.33 ± 0.54	
Yes	4787 (9.8%)	4.55 ± 0.34		1.46 ± 0.64	
PIH	3531 (7.2%)	4.56 ± 0.33		1.44 ± 0.61	
Preeclampsia	1256 (2.6%)	4.53 ± 0.37		1.52 ± 0.70	
ICP			0.039		0.606
No	48,518 (91.1%)	4.45 ± 0.36		1.34 ± 0.55	
Yes	455 (0.9%)	4.41 ± 0.34		1.33 ± 0.50	
Postpartum hemorrhage			0.011		0.000
No	48,237 (98.5%)	4.44 ± 0.36		1.34 ± 0.55	
Yes	736 (1.5%)	4.48 ± 0.36		1.45 ± 0.63	

Categorical data are presented as N (%). Continuous variables are presented as means ± SD.

**Table 2 nutrients-14-03295-t002:** Maternal FPG in early pregnancy categories and the risks of pregnancy outcomes.

Variable	FPG in Early Pregnancy				*p* Value
	<25th	25– <50th	50– <75th	≥75th	
**Total**	12,175	12,181	12,260	12,357	
GDM	993 (8.2%)	1287 (10.6%)	1602 (13.1%)	2773 (22.4%)	0.000
OR (95%CI)	reference	1.33 (1.22–1.45)	1.69 (1.56–1.84)	3.26 (3.02–3.52)	
AOR (95%CI)	reference	1.27 (1.16–1.38)	1.56 (1.44–1.70)	2.81 (2.60–3.05)	
HDP	801 (6.6%)	877 (7.2%)	1461 (11.9%)	1648 (13.3%)	0.000
OR (95%CI)	reference	1.10 (0.99–1.22)	1.92 (1.76–2.10)	2.19 (2.00–2.39)	
AOR (95%CI)	reference	1.08 (0.97–1.19)	1.84 (1.68–2.02)	1.98 (1.81–2.16)	
PIH	558 (4.6%)	611 (5.0%)	1137 (9.3%)	1225 (9.9%)	0.000
OR (95%CI)	reference	1.10 (0.98–1.24)	2.13 (1.92–2.36)	2.29 (2.07–2.54)	
AOR (95%CI)	reference	1.07 (0.95–1.21)	2.03 (1.83–2.26)	2.07 (1.87–2.30)	
Preeclampsia	243 (2.0%)	266 (2.2%)	324 (2.6%)	423 (3.4%)	0.000
OR (95%CI)	reference	1.10 (0.92–1.31)	1.33 (1.13–1.58)	1.74 (1.48–2.04)	
AOR (95%CI)	reference	1.07 (0.90–1.28)	1.28 (1.08–1.52)	1.57 (1.33–1.84)	
ICP	126 (1.0%)	117 (1.0%)	113 (0.9%)	99 (0.8%)	0.282
OR (95%CI)	reference	0.93 (0.72–1.19)	0.89 (0.70–1.15)	0.77 (0.59–1.01)	
AOR (95%CI)	reference	0.93 (0.72–1.20)	0.88 (0.68–1.13)	0.78 (0.60–1.02)	
Postpartum hemorrhage	159 (1.3%)	186 (1.5%)	179 (1.5%)	212 (1.7%)	0.067
OR (95%CI)	reference	1.17 (0.95–1.45)	1.12 (0.90–1.39)	1.32 (1.07–1.62)	
AOR (95%CI)	reference	1.14 (0.92–1.42)	1.08 (0.87–1.34)	1.22 (0.99–1.50)	
Preterm delivery	468 (3.8%)	518 (4.3%)	438 (3.6%)	511 (4.1%)	0.029
OR (95%CI)	reference	1.11 (0.98–1.26)	0.93 (0.81–1.06)	1.08 (0.95–1.23)	
AOR (95%CI)	reference	1.10 (0.97–1.26)	0.90 (0.79–1.03)	1.02 (1.05–1.39)	
SGA	1254 (10.3%)	1212 (9.9%)	1182 (9.6%)	1154 (9.3%)	0.000
OR (95%CI)	reference	0.98 (0.90–1.06)	0.95 (0.88–1.04)	0.94 (0.86–1.02)	
AOR (95%CI)	reference	0.99 (0.91–1.07)	0.97 (0.89–1.05)	0.96 (0.88–1.05)	
LGA	1034 (8.5%)	1192 (9.8%)	1294 (10.6%)	1477 (12.0%)	0.000
OR (95%CI)	reference	1.17 (1.07–1.27)	1.27 (1.16–1.38)	1.45 (1.34–1.58)	
AOR (95%CI)	reference	1.14 (1.04–1.25)	1.21 (1.11–1.32)	1.31 (1.20–1.42)	
Low birth weight (<2500 g)	317 (2.6%)	346 (2.8%)	311 (2.5%)	333 (2.7%)	0.487
OR (95%CI)	reference	1.09 (0.94–1.28)	0.97 (0.83–1.14)	1.04 (0.89–1.21)	
AOR (95%CI)	reference	1.09 (0.94–1.28)	0.98 (0.83–1.14)	1.02 (0.87–1.19)	
Macrosomia (≥4000 g)	559 (4.6%)	636 (5.2%)	726 (5.9%)	860 (7.0%)	0.000
OR (95%CI)	reference	1.15 (1.02–1.29)	1.31 (1.17–1.47)	1.55 (1.39–1.73)	
AOR (95%CI)	reference	1.10 (0.98–1.24)	1.23 (1.10–1.38)	1.35 (1.21–1.52)	
Apgar score ≤7	86 (0.7%)	106 (0.9%)	102 (0.8%)	114 (0.9%)	0.288
OR (95%CI)	reference	1.23 (0.93–1.64)	1.18 (0.88–1.57)	1.31 (0.99–1.73)	
AOR (95%CI)	reference	1.22 (0.92–1.63)	1.15 (0.86–1.53)	1.23 (0.92–1.63)	
NICU	1134 (9.3%)	1280 (10.5%)	1331 (10.9%)	1360 (11.0%)	0.000
OR (95%CI)	reference	1.14 (1.05–1.24)	1.19 (1.09–1.29)	1.20 (1.11–1.31)	
AOR (95%CI)	reference	1.14 (1.05–1.24)	1.19 (1.10–1.30)	1.21 (1.11–1.32)	

Data are shown as N (%), OR (95%CI), AOR (95%CI). In AOR analysis, data are adjusted for pre-pregnancy BMI, years of education, birthplace, parity, maternal age.

**Table 3 nutrients-14-03295-t003:** Associations between mTG in early pregnancy and pregnancy complications stratified by maternal FPG in early pregnancy.

		TG Category Stratified by FPG Categories							
		FPG < 25th		25th ≤ FPG < 50th		50th ≤ FPG < 75th		FPG ≥ 75th	
		mTG < 2.00	mTG ≥ 2.00	mTG < 2.00	mTG ≥ 2.00	mTG < 2.00	mTG ≥ 2.00	mTG < 2.00	mTG ≥ 2.00
GDM	OR AOR	Reference	2.97 (2.50–3.53) 2.53 (2.12–3.02)	Reference	2.51 (2.15–2.95) 2.10 (1.78–2.48)	Reference	2.11 (1.82–2.44) 1.77 (1.52–2.06)	Reference	2.44 (2.18–2.72) 2.07 (1.85–2.32)
HDP	OR AOR	Reference	1.81 (1.46–2.23) 1.57 (1.26–1.96)	Reference	2.03 (1.68–2.46) 1.74 (1.43–2.13)	Reference	1.50 (1.27–1.76) 1.34 (1.13–1.59)	Reference	1.45 (1.27–1.68) 1.28 (1.10–1.48)
PIH	OR AOR	Reference	1.79 (1.39–2.29) 1.51 (1.17–1.96)	Reference	1.73 (1.37–2.20) 1.52 (1.19–1.93)	Reference	1.30 (1.08–1.57) 1.16 (0.96–1.41)	Reference	1.35 (1.15–1.59) 1.20 (1.02–1.42)
Preeclampsia	OR AOR	Reference	1.72 (1.19–2.50) 1.60 (1.08–2.35)	Reference	2.53 (1.85–3.44) 2.09 (1.52–2.89)	Reference	2.04 (1.53–2.72) 1.85 (1.38–2.50)	Reference	1.63 (1.27–2.08) 1.38 (1.07–1.78)

Data are shown as OR (95%CI) and AOR (95%CI). The reference group for the analyses is mother with mTG < 90th (2.00 mM) in each FPG category. In AOR analysis, data was adjusted for pre-pregnancy BMI, years of education, birthplace, parity, and maternal age.

## Data Availability

The data presented in this study are available on request from the corresponding author.

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
