# Peer review of "Effect of Maternal Glucose and Triglyceride Levels during Early Pregnancy on Pregnancy Outcomes: A Retrospective Cohort Study"

_nutrients, 2022, doi:10.3390/nu14163295_

Round 1

Reviewer 1 Report

Some issues to resolve:

1. In tables there is a gradient for some outcome between quartiles; e.g. GDM 8.2%,10.6%,13.1%,22.4%; HDP 6.6%,7.2%,11.9%,13.3%. This is interesting and would deserve some more discussion around significance of this observation.

2. For some outcome (e.g. SGA) lower quartile seems to be associated with a slightly more adverse outcome (SGA 10.3%,9.9%, 9.6%, 9.3%. Is it right? Some more discussion about that would be advisable.

Author Response

Response to Reviewer 1 Comments

Thanks for your professional advice and our responses are as follows:

1.In tables there is a gradient for some outcome between quartiles; e.g. GDM 8.2%,10.6%,13.1%,22.4%; HDP 6.6%,7.2%,11.9%,13.3%. This is interesting and would deserve some more discussion around significance of this observation.

Response:Good suggestion and we added discussion around significance of GDM and HDP incidences between quartiles in the revised manuscript.

2. For some outcome (e.g. SGA) lower quartile seems to be associated with a slightly more adverse outcome (SGA 10.3%,9.9%, 9.6%, 9.3%. Is it right? Some more discussion about that would be advisable.

Response: Good question! According to the recent studies, FPG below the diagnostic threshold and SGA were inversely related, which is in consistency with our study result. We have added more discussion to the revised manuscript.

Reviewer 2 Report

The main merit of the present study is that it is a retrospective cohort study that includes a large number of pregnant mothers. The aim is to evaluate the associations of maternal glucose below diagnostic thresholds and triglyceride during early pregnancy, with the risks of pregnancy complications and adverse perinatal outcomes. The results indicate that screening for glucose and triglycerides in the first weeks of pregnancy can improve treatment and prevent problems in pregnant mothers and their offspring.

The title does accurately reflect the content of the article.

The experimental design and sampling are well explained and reproducible.

The language used is clear and fits the scientific style.

The bibliographic references are adequate and are limited to the last 10 years.

Finally, I would like to raise some questions that I think the authors should take into account:

·       In “Material and Methods”, the city and country where the hospital is located should be indicated.

·       In "Data collection", the acronym BMI appears for the first time and its meaning (body mass index) is not indicated.

·   Indicate in Table 1, with letters whether there are statistically significant differences, as well as the significance level

·        Try to make figure 2 more visible

Author Response

Response to Reviewer 2 Comments

the main merit of the present study is that it is a retrospective cohort study that includes a large number of pregnant mothers. The aim is to evaluate the associations of maternal glucose below diagnostic thresholds and triglyceride during early pregnancy, with the risks of pregnancy complications and adverse perinatal outcomes. The results indicate that screening for glucose and triglycerides in the first weeks of pregnancy can improve treatment and prevent problems in pregnant mothers and their offspring.

The title does accurately reflect the content of the article.

The experimental design and sampling are well explained and reproducible.

The language used is clear and fits the scientific style.

The bibliographic references are adequate and are limited to the last 10 years.

Response: Thanks for your professional comments!

Finally, I would like to raise some questions that I think the authors should take into account:

  • In “Material and Methods”, the city and country where the hospital is located should be indicated.

Response:Thanks for reminding, we added the specific information of the hospital's location in the revised manuscript.

  • In "Data collection", the acronym BMI appears for the first time and its meaning (body mass index) is not indicated.

Response: Thanks for your suggestion, the full spelling of the acronym "BMI" has been added in the "Data collection" part.

  • Indicate in Table 1, with letters whether there are statistically significant differences, as well as the significance level.

Response: Thanks for your wise advice! we added “P value” to Table 1, which has been attached to the revised manuscript and the response, please see the attachment.

  • Try to make figure 2 more visible.

Response: Thanks for your advice,we edited figure 2 to make it more visible, and the new image has been uploaded. Please see the attachment.
